# Three-Dimensional Software- and MR-Imaging-Based Muscle Volumetry Reveals Overestimation of Supraspinatus Muscle Atrophy Using Occupation Ratios in Full-Thickness Tendon Tears

**DOI:** 10.3390/healthcare10101899

**Published:** 2022-09-28

**Authors:** Sophia Samira Goller, Bernd Erber, Nicola Fink, Tobias Rosenkranz, Christian Glaser, Jens Ricke, Andreas Heuck

**Affiliations:** 1Department of Radiology, University Hospital, LMU Munich, 81377 Munich, Germany; 2Division of Cardiovascular Imaging, Department of Radiology and Radiological Science, Medical University of South Carolina, Charleston, SC 29425, USA; 3Radiologisches Zentrum München, Pippinger Str. 25, 81245 Munich, Germany

**Keywords:** supraspinatus tear, muscle atrophy, tendon retraction, occupation ratio, magnetic resonance imaging

## Abstract

Supraspinatus muscle atrophy is widely determined from oblique-sagittal MRI by calculating the occupation ratio. This ex vivo and clinical study aimed to validate the accuracy of 3D software- and MR-imaging-based muscle volumetry, as well as to assess the influence of the tear pattern on the occupation ratio. Ten porcine muscle specimens were volumetrized using the physical water displacement volumetry as a standard of reference. A total of 149 individuals with intact supraspinatus tendons, partial tears, and full-thickness tears had 3T MRI. Two radiologists independently determined occupation ratio values. An excellent correlation with a Pearson’s r of 0.95 for the variables physical volumetry using the water displacement method and MR-imaging-based muscle volumetry using the software was found and formed the standard of reference for the patient study. The inter-reader reliability was 0.92 for occupation ratios. The correlation between occupation ratios and software-based muscle volumes was good in patients with intact tendons (0.84) and partial tears (0.93) but considerably lower in patients with full-thickness tears (0.68). Three-dimensional-software- and MR-imaging-based muscle volumetry is reliable and accurate. Compared to 3D muscle volumetry, the occupation ratio method overestimates supraspinatus muscle atrophy in full-thickness tears, which is most likely due to the medial retraction of the myotendinous unit.

## 1. Introduction

Supraspinatus (SSP) tendon tears are associated with a decreased muscle quality, characterized by muscle atrophy and fatty infiltration [1,2,3,4]. Magnetic resonance imaging (MRI) accurately detects and defines the extent of both partial and full-thickness rotator cuff (RC) tendon tears. However, due to technical limitations of currently available hardware and software, clinical MRI frequently does not cover the SSP muscle in its full medial to lateral extension and therefore does not allow for estimating muscle atrophy by measuring the entire muscle volume [5,6,7]. Instead, the muscle volume and the degree of muscle atrophy, respectively, are widely estimated either by using the qualitative tangent sign described by Zanetti et al. [8] or by assessing the occupation ratio (OR) based on the quotient of the cross-sectional area of the SSP muscle and that of the supraspinatus fossa from an oblique-sagittal image on the level of the Y-view, as first described by Thomazeau et al. [9]. However, both methods provide only a two-dimensional (2D) representation of the three-dimensional (3D) muscle volume and both have been shown to be less valid than 3D muscle volume analysis in previous studies [5,7,10]. Therefore, the OR method might not reliably represent the SSP muscle volume, particularly in cases of full-thickness tendon tears, as it may be influenced by the medial retraction of the myotendinous unit. Although controversially discussed at some point, several previous studies suggest that significant degrees of SSP muscle atrophy are associated with a limited prognosis after RC repair procedures; therefore, the preoperative assessment of muscle atrophy is crucial for orthopedic surgeons in evaluating treatment indications [11,12,13,14]. With this background, this study had two major goals: first, to validate the accuracy of 3D software- and MR-imaging-based SSP muscle volumetry in an ex vivo setting, and, second, to assess the influence of the SSP tear pattern (partial vs. full-thickness tear) on the OR when compared to 3D muscle volumetry. It was hypothesized that SSP muscle atrophy would be overestimated in full-thickness tendon tears due to the above-mentioned medial tendon retraction effect.

## 2. Materials and Methods

This study consisted of two parts. First, an experimental ex vivo study was performed in order to validate the accuracy of the 3D software- and MR-imaging-based SSP muscle volumetry measurement method, which was used later on in the patient study as a standard of reference. Subgroup analyses were performed within the patient cohort, evaluating intact SSP tendons as well as partial and, respectively, full-thickness SSP tendon tears.

### 2.1. Ex Vivo Study

The accuracy of 3D muscle volumetry based on T1-weighted paracoronal MR sequences and using the mintLesion^TM^ software (Version 3.4.5, Mint Medical GmbH, Heidelberg, Germany) was not proven in previous studies. Therefore, ten porcine muscle specimens were volumetrized in an ex vivo setting first. All muscle specimens had a spindle-shaped configuration, which was similar to that of the human SSP muscle. In order to exactly define the volume of these muscle specimens, the physical water displacement method was used, serving as the standard of reference. In detail, to determine the volumes of the muscle specimens, a borosilicate glass measuring cylinder with a nominal volume of 100 cm^3^ and a measurement accuracy of ±2 cm^3^ was used. Muscle specimen volumes varied between 10 and 100 cm^3^. The next step was to generate paracoronal T1-weighted TSE (turbo spin-echo) sequences of the porcine muscle specimens with the same 3T unit as that used for patients’ examinations. Thereafter, the volumes of the porcine muscle specimens were semiautomatically determined from each image data set by using the mintLesion^TM^ software (Figure 1). This was carried out by two radiologists independently analogous to the SSP muscle volume measurements in the patient study as described below.

### 2.2. MR Imaging

All MR imaging acquisitions, including the ex vivo and patient study, were conducted on a 3T scanner (MAGNETOM Skyra, Siemens Healthineers, Erlangen, Germany) with a dedicated 16-channel phased-array shoulder coil. The imaging protocol of the ex-vivo study consisted of a T1-weighted TSE sequence with a field of view (FOV) of 200 mm^2^, TR/TE of 600/8.9 ms, and a slice thickness of 3 mm, with a spacing of 0.3 mm. The imaging protocol for the patient study included a total of four sequences, all with 3 mm slice thickness and 0.3 mm spacing: (1) a paracoronal T1-weighted TSE sequence, (2) paracoronal and (3) axial fat-suppressed intermediate-weighted multishot TSE sequences, and (4) a parasagittal T2-weighted multishot TSE sequence. Detailed imaging parameters are listed in Table 1.

### 2.3. Patient Study

A total of 149 individuals who underwent 3T MRI of the shoulder between 1 September and 30 November 2020, for clinical indications such as shoulder impingement, glenohumeral or acromioclavicular instability, and suspected adhesive capsulitis, were included. Patients’ characteristics are summarized in Table 2. Patients were retrospectively identified via a full-text query within the local radiology information system using the search term “supraspinatus”. Subsequently, the resulting studies were further filtered concerning the availability of fully covered SSP muscles from their medial to lateral extents on paracoronal T1-weighted sequences. Individuals with associated or competing pathologies, such as advanced osteoarthritis ≥ grade II according to the extended classification by Samilson–Habermeyer [15], a previous shoulder fracture, or any RC tears other than the SSP tendon, were excluded. The study cohort was divided into individuals with intact SSP tendons, individuals with partial SSP tendon tears, and those with full-thickness SSP tendon tears (symptom duration ≥ 3 months) by a senior musculoskeletal radiologist with more than 30 years of experience (AH), respectively.

### 2.4. MRI Evaluation

For all subsequent evaluations, the anonymized MRI data were exported from the local picture archiving and communicating system (PACS) and uploaded into the mintLesion^TM^ software.

### 2.5. Three-Dimensional Supraspinatus Muscle Volume Measurement

For 3D SSP muscle volume measurements, one radiologist (SSG) manually segmented each SSP muscle by tracing its margins on paracoronal T1-weighted images slice by slice using the mintLesion^TM^ software (Figure 2). The largest axial diameter (long axis) and the measurement perpendicular to the long axis (short axis) were automatically derived by the software from the outlined muscle areas. The software then automatically calculated cross-products (CPs) from the long and short muscle axes on each slice. From all outlined SSP muscle areas and the slice thickness (including interslice gap), the muscle volume was automatically derived by the software.

### 2.6. Occupation Ratio Evaluation

ORs were determined by two observers (SSG, BE) independently, both of whom were blinded to clinical data. The OR of the supraspinatus fossa according to Thomazeau et al. [9] was calculated by building the quotient of the cross-sectional surface area of the SSP muscle and the supraspinatus fossa area from an oblique-sagittal T2-weighted sequence (Figure 3). According to the initial study by Thomazeau et al., the SSP muscle was considered normal or mildly atrophied with an OR between 1.00 and 0.60 (stage I). In contrast, OR values between 0.60 and 0.40 (stage II) indicate moderate atrophy, and values below 0.40 (stage III) suggest severe atrophy [9]. The single oblique-sagittal slice used for the measurement was selected, where a “Y” is formed by the bony landmarks of the scapular spine, the coracoid process, and the distal clavicle, known as the Y-view [9]. Each SSP muscle and supraspinatus fossa was manually segmented by tracing its margins in the mintLesion^TM^ software. From the outlined areas of the SSP muscle and the supraspinatus fossa, respectively, the largest axial diameter (long axis) and the measurement perpendicular to the long axis (short axis) were automatically derived. Then, CPs were calculated by the software by multiplying the long and short axes of the segmented areas.

### 2.7. Statistical Analysis

Statistical analysis was performed using R-Studio (Version 4.0.4, packages “FSA” and “DescTools” RStudio Inc., Boston, MA, USA). Inter-reader reliability for ORs and correlation between ORs and software-determined 3D muscle volumes were calculated with Pearson’s correlation coefficient. Measurements of mean muscle volumes among the groups of intact SSP tendons, partial SSP tendon tears, and full-thickness SSP tendon tears were analyzed using a one-way analysis of variance (ANOVA). Bonferroni post hoc analysis was applied. A comparison between OR values and 3D muscle volumes for full-thickness tendon tears was carried out in relation to equal measurements for intact tendons, respectively, as units for both measurements differ. A *p*-value of 0.05 was set as the limit of statistical significance.

The institutional research committee approved this study (approval number 20-814). Informed consent was not required because of a retrospective study design and anonymization of patients’ data.

## 3. Results

### 3.1. Ex Vivo Study

In the ex vivo study of porcine muscle specimens, an excellent correlation with a Pearson’s r of 0.95 for the variables physical volumetry using the water displacement method and MR-imaging-based 3D SSP muscle volumetry using the mintLesion^TM^ software based on the results of two independent readers was found. This allows software-based volumetry to serve as the standard of reference for the subsequent patient study. The results of the ex vivo study are summarized in Table 3.

### 3.2. Patient Study

#### 3.2.1. SSP Muscle Volumes

Software-derived 3D SSP mean muscle volumes determined from paracoronal T1-weighted sequences were highest in the intact SSP tendon group (mean volume 56.9 cm^3^), lower in the partial tendon tear group (mean volume 43.6 cm^3^; *p* < 0.001), and again lower in the subgroup of patients with full-thickness SSP tendon tears (mean volume 31.1 cm^3^; *p* < 0.001) (Figure 4). A subgroup analysis within the full-thickness SSP tendon tear group evaluating tendon retraction grades 1, 2, and 3 according to the Patte classification system [16] showed no significant differences in mean muscle volumes.

#### 3.2.2. Supraspinatus Fossa OR Values

The inter-reader reliability was excellent, with 0.92 for OR values. The results for supraspinatus fossa OR values in the patient cohort according to the Thomazeau classification are summarized in Table 4.

#### 3.2.3. Correlation between 3D SSP Muscle Volumes and OR Values

The correlation between 3D muscle volumes and OR values was good for intact SSP tendons (0.84) and partial SSP tendon tears (0.93) but considerably lower (0.68) for full-thickness SSP tendon tears with any degree of medial retraction of the myotendinous unit (Figure 5). In the full-thickness SSP tendon tear subgroup, the OR method overestimated muscle atrophy compared to MR-imaging-based 3D muscle volumes, as determined with the *mintLesion^TM^* software, by 13.7% in relation to measurements for intact SSP tendons (Figure 5; relative SSP muscle volume of full-thickness tears compared to intact SSP tendons using MR-imaging-based 3D muscle volumes: 54.7%; SSP Thomazeau muscle surface of full-thickness tears compared to intact SSP tendons: 47.2%; 54.7% − 47.2% = 7.5%; 0.075/0.547 = 0.137).

## 4. Discussion

The major finding of this study was that the degree of SSP muscle atrophy is overestimated on the commonly used Y-view in full-thickness tears with any degree of myotendinous unit retraction when compared to 3D muscle volumetry. However, this study found a good correlation between OR values and muscle volumes in the subgroups of intact SSP tendons and partial SSP tendon tears, respectively.

Our data demonstrate that 3D muscle volumetry should be preferred in patients with full-thickness SSP tendon tears when the muscle volume needs to be quantified preoperatively in order to estimate the postoperative success rate after tendon repair procedures. This is in accordance with previous studies [17] and is important since various previous experimental and clinical studies have shown that significant SSP muscle atrophy is an important adverse prognostic factor affecting patients’ outcomes in addition to other conditions, e.g., fatty infiltration or severe tear size [8,18,19,20,21,22,23,24].

Over the past decades, MRI has been established as the pretherapeutic standard method of choice, both for making the diagnosis of RC tendon tears as well as for the assessment of muscle atrophy and fatty infiltration, which is essential in surgical decision-making [4,5,7,8,9,25]. Both qualitative and quantitative measurement methods for determining the degree of SSP muscle atrophy were integrated into clinical practice. The qualitative tangent sign, which was first described by Zanetti et al. [8], and the quantitative OR method, which was first described by Thomazeau et al. [9], became the most commonly used tools to estimate muscle atrophy in the pretherapeutic work-up of patients with SSP pathologies. The Y-view, which is defined as the most lateral oblique-sagittal image slice on which the scapular spine is in contact with the coracoid process, was introduced to be the most favorable view for a reliable estimation of SSP muscle atrophy with a high rate of reproducibility [9]. Thereby, the last aspect mentioned is in accordance with the findings in our study. The inter-reader reliability in determining the OR on the Y-view was high (0.92), which is similar to the initial study of Thomazeau et al., who reported a positive correlation coefficient of ≥0.90 for three readers calculating the OR in 55 patients [9]. In line with these findings, Fukuta et al. reported an inter-reader reliability for two observers determining the cross-sectional area of the SSP muscle and SSP fossa in 76 shoulders of 0.995 and 0.959, respectively [26]. This is probably due to the concise bony landmarks of the Y-view.

However, it has not yet been securely proven whether the 2D OR value determined on the Y-view position from just one single oblique-sagittal slice is representative of the total SSP muscle volume, which represents a 3D geometric figure. Instead, it is rather suspicious that the OR value might be diminished in full-thickness SSP tendon tears due to the medial retraction of the whole myotendinous unit when compared to 3D muscle volumetry. This is what we were able to show in our study: the OR determined on the Y-view decreased with an increasing severity of the SSP tear pattern. While the mean OR value in the intact SSP tendon group was 0.72, it was 0.54 in the partial tendon tear group and 0.40 in the full-thickness tendon tear group, respectively. These results are in line with those of previous studies [8,9,26]. In line with this assumption, previous results of Fukuta et al. already suggested the hypothesis that a lateral located single oblique-sagittal slice might not be representative of the entire SSP muscle volume in full-thickness tears [26]. Their results showed that the OR in medium size tears (sagittal extent of the tear of 1–3 cm according to the classification of Cofield et al. [27]) was <0.5 on the Y-view in more than half of their cases, which indicates SSP muscle atrophy. However, the OR on three more medial slices than the Y-view was higher in their cohort and similar to that of their impingement syndrome control group. Interestingly, a significant decrease in the cross-sectional area of the SSP muscle on the next three slices medial to the Y-view was found by Fukuta et al. only in patients with large and massive tears [26]. As a limitation, with their approach, they were not able to determine the total SSP muscle volume three-dimensionally and the most appropriate slice position medial to the Y-view on oblique-sagittal MR images remained unclear. Nevertheless, in conclusion with previous studies as well as our findings, it should be considered that the main reason for the overestimation of SSP muscle atrophy in full-thickness tears on the standardized Y-view is most likely due to a medialization of the myotendinous unit that influences the cross-sectional surface area of the SSP muscle. Strengthening this hypothesis, Chung et al. confirmed the lateral migration effect of the SSP muscle after RC repair procedures by comparing the preoperative and immediate postoperative total muscle volumes in comparison to the OR values [5]. Similar results with respect to the lateral migration effect were also reported by Jo et al., who assessed immediate postoperative changes after arthroscopic RC repair by comparing preoperative MRI studies with those undertaken three days after surgery. They found that the cross-sectional areas of both the SSP as well as the infraspinatus muscle increased by 21.6% and 7.0%, respectively, and concluded that a lateral shift of the muscle bellies is caused by bringing the torn and retracted tendons back to their insertion zone on the greater tuberosity, which might alter the appearance of the muscle bellies on the Y-view, leading to an increase in cross-sectional surface area values [28].

This study was the first to prove an overestimation of SSP muscle atrophy in full-thickness tears by using the OR method when 3D software- and MR-imaging-based volumetry serves as the standard of reference. In addition, our study strengthens the results of previous studies, which already suspected an overestimation of SSP muscle atrophy in full-thickness tendon tears on the Y-view as a result of medial retraction of the myotendinous unit.

Nevertheless, our study had some potential limitations. First, the number of patients with intact SSP tendons and full-thickness SSP tendon tears is relatively small when compared to the number of individuals with partial tendon tears according to Ellman, which, to some degree, reflects the frequency of these entities among patients that are referred to an MRI of the shoulder but might potentially have led to some risk of selection bias. Second, the human SSP muscle volumes were segmented by only one single observer, which might lead to an investigator bias. Third, changes in OR values on more medial slices compared to the conventional Y-view were not evaluated, though it has already been shown that this enables a more accurate estimation of the SSP muscle volume in former studies [5,26].

Based on the results of this study, 3D muscle volumetry provides more accurate results in defining the extent of SSP muscle atrophy in full-thickness tendon tears than the standardized 2D OR approach. Beyond this, software- and MR-imaging-based 3D SSP muscle volumetry has proven its accuracy in an ex vivo pre-study and may potentially be applied in further investigations related to the assessment of RC muscle volumes.

## 5. Conclusions

In the ex vivo part, this study demonstrated that 3D-software- and MR-imaging-based muscle volumetry is reliable and accurate. The application of this technique to the SSP muscle in individuals with and without tendon tears and a comparison with respective ORs revealed that the OR method overestimates SSP muscle atrophy in full-thickness tendon tears. This is most likely due to the medial retraction of the myotendinous unit in fully torn tendons and should be considered when MRI studies of patients with full-thickness SSP tears are assessed for muscle atrophy in the preoperative setting. Including a T1-weighted paracoronal sequence to standard clinical shoulder protocols that entirely covers the SSP muscle in its medial to lateral, as well as its anterior to posterior extension, thus allowing us to determine the SSP muscle volume three-dimensionally with clinical available tools, should be considered.

## Figures and Tables

**Figure 1 healthcare-10-01899-f001:**
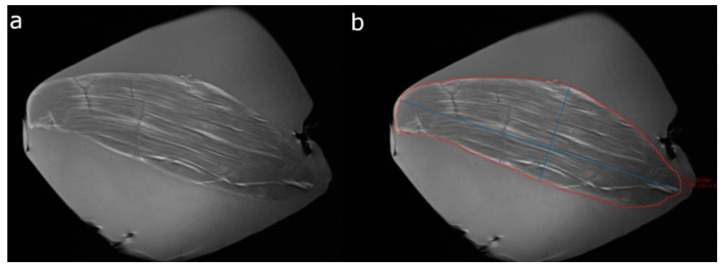
Sample case demonstrating semiautomated volume measurement of a porcine muscle specimen. (**a**) Paracoronal T1-weighted image of a porcine muscle specimen with a volume of 90 cm^3^ as previously determined by the physical water displacement method. (**b**) Muscle specimen margins are manually traced (red outline) on each image slice of the data set using the mint Lesion^TM^ software and automated 2D measurements (largest long axis dimension and perpendicular short axis dimension, blue lines) are automatically derived for each slice. The volume of the whole muscle specimen is then automatically calculated by the software.

**Figure 2 healthcare-10-01899-f002:**
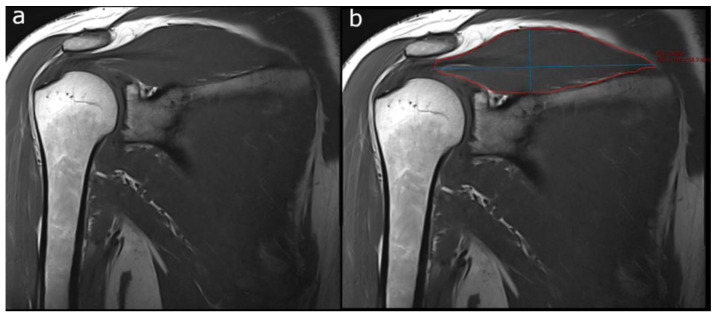
Sample case demonstrating semiautomated muscle volume assessment with the mint Lesion^TM^ software. (**a**) Paracoronal T1-weighted image of the right shoulder. (**b**) SSP muscle margins are traced (red outline) on each image slice and automated 2D measurements (largest long axis dimension and perpendicular short axis dimension, blue lines) are automatically derived for each slice, forming the base for automated 3D volume calculation of the whole SSP muscle by the software.

**Figure 3 healthcare-10-01899-f003:**
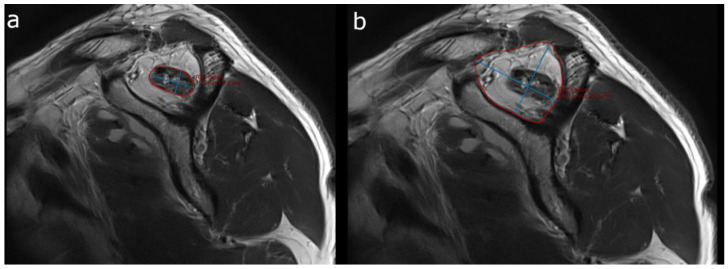
Sample case demonstrating semiautomated OR-value measurements. (**a**) Margins of the SSP muscle belly and (**b**) the supraspinatus fossa are manually traced (red outline) on the Y-view (oblique-sagittal T2-weighted sequence), and automated 2D measurements (largest long axis dimension and perpendicular short axis dimension, blue lines) are automatically derived. Cross-sectional areas are calculated by multiplying the long and short axes in the mint Lesion^TM^ software.

**Figure 4 healthcare-10-01899-f004:**
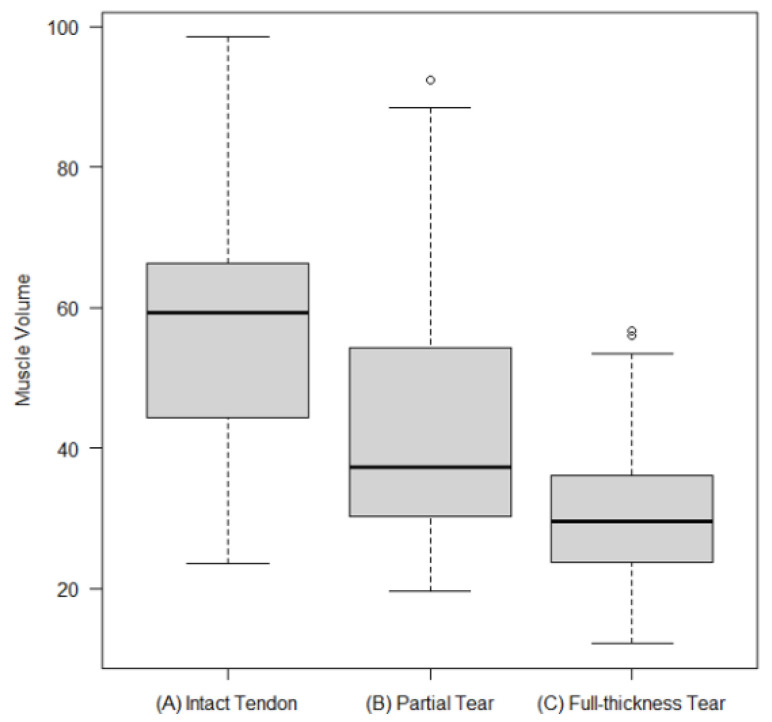
Boxplots showing minimum, first quartile, median, third quartile, and maximum for software-determined muscle volumes in the (**A**) intact SSP tendon subgroup (mean volume 56.9 cm^3^), (**B**) partial SSP tendon tear subgroup (mean volume 43.6 cm^3^), and (**C**) full-thickness SSP tendon tear subgroup (mean volume 31.1 cm^3^; *p* < 0.001).

**Figure 5 healthcare-10-01899-f005:**
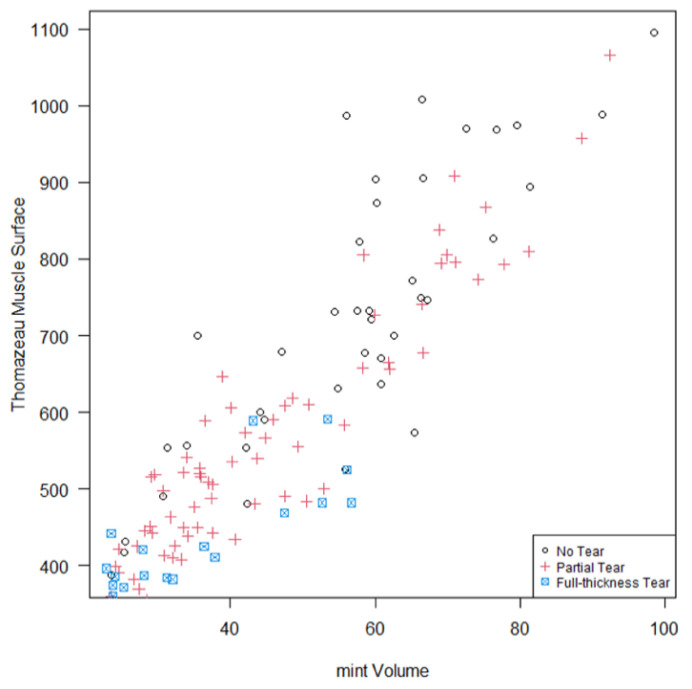
The correlation between OR values and muscle volumes was high for intact SSP tendons (0.84) and partial SSP tendon tears (0.93) but considerably reduced for full-thickness SSP tendon tears with medial retraction of the myotendinous unit (0.68).

**Table 1 healthcare-10-01899-t001:** Magnetic resonance imaging parameters.

Parameter	Paracoronal T1(w) TSE	Paracoronal fs Intermediate (w) Multishot TSE	Axial fs Intermediate (w) Multishot TSE	Parasagittal T2 (w) Multishot TSE
TR [ms]	600	3000	3500	5370
TE [ms]	8.9	44	46	89
Slices [n]	23	23	36	32
Slice thickness [mm]	3	3	3	3
Spacing [mm]	0.3	0.3	0.3	0.3
Matrix	384 × 384	384 × 384	320 × 320	320 × 320
FOV [mm^2^]	200	150	160	140
TA [min:s]	1:34	2:57	3:02	2:49

Fs, fat-suppressed; FOV, field of view; TA, acquisition time; TE, echo time; TR, repetition time; TSE, turbo spin-echo; (w), weighted.

**Table 2 healthcare-10-01899-t002:** Patients’ characteristics.

Characteristics	Study Cohort
Number of individuals	149
Age in years ± SD (range)	55.0 ± 16.0 (19–85)
Gender, *n* (%)	
Male	83 (55.7)
Female	66 (44.3)
Side, *n* (%)	
Right	69 (46.3)
Left	80 (53.6)
SSP tendon, *n* (%)	
Intact	39 (26.2)
Partial tear	75 (50.3)
Grade I *	45
Grade II *	19
Grade III *	11
Full-thickness tear	35 (23.5)
Grade I **	19
Grade II **	10
Grade III **	6

SSP, supraspinatus; * according to the Ellman classification [15]: incomplete supraspinatus tear in the articular or bursal surface with a tear depth of less than 3 mm respectively involving < ¼ of the tendon diameter (grade I), a tear depth of 3–6 mm, respectively, involving < ½ of the tendon diameter (grade II), or a tear depth of >6 mm, respectively, involving > ½ of the tendon diameter (grade III); ** according to the Patte classification [16]: grade 1 describes a full-thickness SSP tendon tear with the proximal tendon stump near the bony insertion, grade 2 is consistent with a full-thickness SSP tendon tear with the proximal tendon stump at the level of the dome of the humeral head, and grade 3 is defined as a full-thickness SSP tendon tear with the tendon stump at the glenoid level or more proximal.

**Table 3 healthcare-10-01899-t003:** Results for muscle specimen volumes derived by water displacement volumetry and software-based volumetry.

Muscle Specimen [*n*]	Water Displacement Volumetry [mL]	Software-Based Volumetry [mL]	%
1	10	9.97	99.70
2	20	19.48	97.40
3	30	30.29	100.97
4	40	39.68	99.20
5	50	49.81	99.62
6	60	61.62	102.70
7	70	69.25	98.93
8	80	80.34	100.43
9	90	89.71	99.68
10	100	99.88	99.88

**Table 4 healthcare-10-01899-t004:** Supraspinatus fossa OR values in the patient cohort.

SSP Tendon Groups [*n*]	OR Value [Mean]	OR Stage 1 * [*n*; %]	OR Stage 2 * [*n*; %]	OR Stage 3 * [*n*; %]
Intact tendon (39)	0.72 (0.49–0.89)	34 (87.2)	5 (12.8)	0 (0)
Partial tear (75)	0.54 (0.26–0.79)	31 (41.3)	38 (50.7)	6 (8.0)
Full-thickness tear (35)	0.40 (0.11–0.71)	1 (2.9)	16 (45.7)	18 (51.4)

SSP, supraspinatus; OR, occupation ratio; * according to the Thomazeau classification [9]: the SSP muscle is considered normal or mildly atrophied with an OR between 1.00 and 0.60 (stage I). In contrast, OR values between 0.60 and 0.40 (stage II) indicate moderate atrophy, and values below 0.40 (stage III) suggest severe atrophy.

## Data Availability

The data presented in this study are available on reasonable request from the corresponding author.

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
