# Peer review of "Three-Dimensional Software- and MR-Imaging-Based Muscle Volumetry Reveals Overestimation of Supraspinatus Muscle Atrophy Using Occupation Ratios in Full-Thickness Tendon Tears"

_healthcare, 2022, doi:10.3390/healthcare10101899_

Round 1
Reviewer 1 Report
The authors are commended on their efforts in (1) validating the accuracy of 3D software- and MR-imaging-based supraspinatus muscle volumetry in an ex-vivo setting, and (2) in assessing the influence of the supraspinatus tear pattern (partial vs. full-thickness tear) on the occupation ratio when compared to 3D muscle volumetry. The authors focused on a very important topic such is supraspinatus muscle atrophy and its correlation with prognosis after rotator cuff repair with several studies in the literature reporting that significant degrees of SSP muscle atrophy are associated with a limited prognosis after RC repair procedures. Therefore, the preoperative assessment of muscle atrophy should be considered crucial for orthopedic surgeons in evaluating treatment indications. Introduction is a concise summary of the literature with appropriate references and it adequately identifies the controversy. Methods are appropriate, stand alone and are reproducible. Results and Discussion are appropriate.
The paper needs only minor revisions limited to text editing.
page 3, line 99: The information regarding the 3T MRI should be moved to line 82.
Author Response
Thank you for your comments. The information regarding the 3T MRI was moved from line 99 to line 82.
Reviewer 2 Report
Dear authors,
The rationale of this study is logical and scientifically sound. However, the result for validating the accuracy of 3T MRI in ex-vivo study is missing, which is an essential part of the study.
Please provide your revision for the comments as the following:
1. Line 63: Please provide the study result of the ex-vivo study
2. Line 185: i) Typo; ii) please elaborate how you deduce the number (highlighted)
3. Line 213: Please describe the meaning of notions in Figure 4
4. Line 218: Please correct the annotation issue
5. Line 225: Please keep the consistency of the description
6. Line 278: Please rephrase the sentence for clarification
7. Line 305: Typo
Please refer to comments in the attached file for suggestions.

Reviewer 3 Report
This paper aims to validate the use of 3D volumetry of SSP and study the effect of tear pattern on the occupation OR compared to the 3D volumetry. These objectives were clear and the conclusions of this paper were adequately supported by the results. There are few minor concerns that must be addressed.
The ex vivo part is very interesting and creative. I wonder if the animal study was approved by the IACUC?.
I noticed that the age range of the subjects was high (19-85 years). Was there a correlation between SSP volume/atrophy and age? The results presented in Figure 4 and Figure 5 may be related to age too.
The sentence in section 3.1 is very long and difficult to read. Please rephrase and break it down into sentences.
Please split the discussion into paragraphs to something like this 1) discussion of the findings with emphasis on its novelty 2) comparison with previous studies and 3) limitations and further studies. No need to stick to the format I suggest but please divide the discussion into several paragraphs, each having its own emphasis.
Round 2
Reviewer 2 Report
Dear authors,
Revisions were appreciated. Please read carefully of both attached documents for more revisions. Please elaborate the result of how you deduce the number of 47.2% for the ratio of OR values, which is the only and crucial result to support your statement.
